# Critical care work during COVID-19: a qualitative study of staff experiences in the UK

Catherine M Montgomery ![ORCID],[1] Sally Humphreys ![ORCID],[2] Corrienne McCulloch,[3] Annemarie B Docherty,[3,4] Steve Sturdy ![ORCID],[1,5] Natalie Pattison ![ORCID] [6,7]

[1]Centre for Biomedicine, Self and Society, University of Edinburgh, Edinburgh, UK
[2]Critical Care and Research & Development, West Suffolk NHS Foundation Trust, Suffolk, UK
[3]Anaesthetics, Theatres and Critical Care, NHS Lothian, Edinburgh, UK
[4]Centre for Medical Informatics, The Usher Institute, University of Edinburgh, Edinburgh, UK
[5]Science, Technology and Innovation Studies, University of Edinburgh, Edinburgh, UK
[6]School of Health and Social Work, University of Hertfordshire, Hertfordshire, UK
[7]Nursing, East and North Hertfordshire NHS Trust, Stevenage, UK

**Correspondence to**
Dr Catherine M Montgomery;
Catherine.montgomery@ed.ac.uk

## ABSTRACT

**Objective** To understand National Health Service (NHS) staff experiences of working in critical care during the first wave of the COVID-19 pandemic in the UK.

**Design** Qualitative study using semistructured telephone interviews and rapid analysis, interpreted using Baehr's sociological lens of 'communities of fate'.

**Participants** Forty NHS staff working in critical care, including 21 nurses, 10 doctors and advanced critical care practitioners, 4 allied health professionals, 3 operating department practitioners and 2 ward clerks. Participants were interviewed between August and October 2020; we purposefully sought the experiences of trained and experienced critical care staff and those who were redeployed.

**Setting** Four hospitals in the UK.

**Results** COVID-19 presented staff with a situation of extreme stress, duress and social emergency, leading to a shared set of experiences which we have characterised as a community of fate. This involved not only fear and dread of working in critical care, but also a collective sense of duty and vocation. Caring for patients and families involved changes to usual ways of working, revolving around: reorganisation of space and personnel, personal protective equipment, lack of evidence for treating COVID-19, inability for families to be physically present, and the trauma of witnessing extreme patient acuity and death on a large scale. The stress and isolation of working in critical care during COVID-19 was mitigated by strong teamwork, camaraderie, pride and fulfilment.

**Conclusion** COVID-19 has changed working practices in critical care and profoundly affected staff physically, mentally and emotionally. Attention needs to be paid to the social and organisational conditions in which individuals work, addressing both practical resourcing and the interpersonal dynamics of critical care provision.

## INTRODUCTION

COVID-19 has placed unprecedented demands on the UK National Health Service (NHS). Around 8% of all hospital admissions, over 14 000 patients, have been admitted to critical care services with COVID-19 since February 2020.[1] [2]Critical care services were rapidly expanded to meet these demands.[3] Research from China and France into the experiences of healthcare staff demonstrates the enormous

### Strengths and limitations of this study

► This is the first study to provide a sociological analysis of critical care work during the first wave of the COVID-19 pandemic in the UK.

► International studies of staff experience of COVID-19 have focused on individualised mental health outcomes; we use the theoretical concept of 'communities of fate' to add value to existing approaches.

► Our sample included a range of professional groups and explicitly sought to capture the experiences of both experienced and redeployed staff.

► Our sample was limited due to the fact that participants were self-selecting and came from a small number of sites.

► Medical and nursing staff made up the majority of participants in our sample; our findings may over-represent the experiences of these professional groups thus limiting wider generalisability.

pressure COVID-19 has placed on doctors and nurses, ranging from issues of healthcare service organisation to personal mental health and well-being.[4 5] In the UK context, Vindrola-Padros *et al* report on anxiety and distress caused by limited personal protective equipment (PPE) for healthcare workers, lack of routine testing and unmet training needs for redeployed staff.[6]

Research on healthcare staff during the pandemic has predominantly focused on the psychological impacts of working in critical care during this time. A European-wide study identified high levels of self-reported burnout (51%) during the pandemic among intensive care unit (ICU) staff respondents.[5] We know that burnout, moral injury and moral distress are significant issues in critical care staff, and prior to COVID-19 were already the subject of a call to arms in the international critical care community.[7] Prevalence in critical care staff ranges from 6% to 47%, with worse burnout compared with areas such as palliative care, and significant emotional labour associated with working in critical care.[8–10]

While much of the research to date has taken a narrowly psychological approach to staff experience, focusing on poor mental health outcomes such as burnout, we aimed to adopt a broader, sociological lens. As Matthewmann and Huppatz note, "As the discipline charged with making sense of contemporary social cohesion and transformation, sociology is well placed to comment on coronavirus and its profound consequences."[11] The sociology of pandemics draws attention to the way in which social institutions—including healthcare systems—change when biological environments change and threaten established ways of living and acting in the world.[12] Social fragilities and structural deficiencies are laid bare and para-epidemics—of fear, of explanation and moralisation, and of action—proliferate.[13 14] In relation to the impacts of COVID-19 on mental health, members of the Society and Mental Health COVID-19 Expert Group recently argued against the pathologisation of responses to the pandemic and instead called for attention to the social substrates of poor mental health.[15] This view is echoed by a recent paper by Vera San Juan *et al*, emphasising the importance of socio-ecological approaches to healthcare worker well-being during the COVID-19 pandemic.[16] We accordingly aimed to provide an alternative to the discourse of individual psychological responses in healthcare workers by exploring issues including changing work organisation and its impacts, identity and care work, and interpersonal/professional relationships.

We draw on the sociological concept 'community of fate' to illuminate the experiences of frontline staff working in critical care during the first wave of the COVID-19 pandemic in the UK. Analysing the 2003 SARS outbreak in Hong Kong, Baehr describes a 'community of fate' as a particular form of group cohesion arising from a situation of extreme duress (figure 1).[17 18] Such communities are socially productive; in the face of an existential threat, they mobilise around shared purpose and resources (including organisation and leadership) to instantiate collective action. A key hallmark of such communities is 'a common focus of sustained attention, and an intense feeling of horizontal interconnectedness'.[18]

## METHODS

We conducted qualitative research using semistructured telephone interviews and rapid analysis.[19 20] Qualitative research can contribute to the evidence base of managing COVID-19 by accessing *how* frontline staff manage their day-to-day work, *why* particular approaches work or not from the point of view of those implementing them, and *what* could be done to improve the experience of caring for patients and families in critical care.[21] Critical care is used as a term throughout to encompass intensive care/ICUs, intensive therapy units and critical care/critical care units.

### Sampling and recruitment

Using principles of maximum variation sampling,[22] we recruited 40 frontline staff members working in critical care from four hospitals in the UK, including nurses, medical staff, allied health professionals (AHPs) and ward clerks. The hospitals were all located in urban areas and served populations of between 500 000 and 2 000 000 inhabitants. At the time of the study, hospital A had more than 25 critical care beds, hospitals B and D had between 15 and 20 critical care beds, and hospital C had fewer than 10 critical care beds. All increased their capacity to deal with the COVID-19 surge in the first wave of the pandemic. We recruited 18 participants from hospital A, 6 from each of hospitals B and C, and 10 from hospital D. The sample was diverse with respect to age, gender and experience working in critical care (table 1). The study was advertised using posters, email and word of mouth. Snowball sampling, which takes advantage of the social networks of identified respondents, provided the research team with an escalating set of participants.

---

**Danger recognition**: Awareness of a pressing and immediate danger, which poses a risk to life and common existence

**Moral density**: Recognition that one's own fate is tie up with others', leading to a sense of social interconnectedness, which may manifest as altruism

**Trial**: The experience of sustained ordeal

**Closure**: Literal and metaphorical curtailment of movement; having to stand one's ground, leading to a sense of collective exile and isolation

**Material and organisational resources**: Resources which help to combat the threat

**Axis of convergence**: May include a common language or civic pride, through which the sense of community is affirmed

**Social ritual**: Actions which provide the group with a specific, crisis-related social identity, which separates them from normal life and provides a basis for social cohesion

**Figure 1** Key features of communities of fate (adapted from Baehr, 2005).[17]

**Table 1** Sample description

| Profession | Redeployed | Female | Male | Total |
|---|---|---|---|---|
| Advanced critical care practitioner | Yes | – | – | – |
| | No | 1 | – | 1 |
| Dietitian | Yes | 1 | – | 1 |
| | No | – | – | – |
| Doctor | Yes | – | 1 | 1 |
| | No | 3 | 5 | 8 |
| Nurse | Yes | 11 | 2 | 13 |
| | No | 8 | – | 8 |
| Operating department practitioner | Yes | 2 | 1 | 3 |
| | No | – | – | – |
| Physiotherapist | Yes | 3 | – | 3 |
| | No | – | – | – |
| Ward clerk | Yes | 2 | – | 2 |
| | No | – | – | – |
| **Total** | | **31** | **9** | **40** |

Interested participants were provided with information about the study, contacted the principal investigator and were subsequently provided with a participant information sheet. Once participants had agreed, verbal informed consent was digitally recorded prior to interview.

### Data collection and analysis
Telephone interviews were conducted between August and October 2020 by CM (sociologist), SH (research nurse/scholar) and NP (clinical professor of nursing). The research team included the additional expertise of a critical care consultant, professor of sociology and nurse researcher in critical care, with significant combined qualitative research experience. Interviews were digitally recorded and professionally transcribed. Interviews lasted from between approximately 30 and 80 min. Semistructured interviews covered staff's experiences of working in critical care during the first wave of the pandemic (roughly between March and July 2020). Questions related to changes in working practice, interaction with patients, technology for family communication, end of life, learning and training, and personal wellbeing and support.

Data were analysed by team members following the rapid analysis methods proposed by Hamilton[19] and elaborated by Taylor *et al*.[20] In the first stage, key issues were noted on a structured summary template describing: participant and data collection details, deductive and inductive headings, quotations and the analyst's reflections. Deductive aspects of the summary template were developed from the research questions. Following an initial testing period, additional, inductively generated subheadings were added. Summarised data were then transferred to a matrix to 'streamline the process of noting simultaneously and systematically similarities, differences and trends in responses across groups of informants'.[23]

Transparent team review and discussion across all transcripts took place to enhance confirmability, trustworthiness, dependability and credibility.[24] Early findings were discussed within the team and subsequently interpreted using the sociological lens of 'communities of fate'.[17]

### Patient and public involvement
Patients and/or the public were not involved in the design, conduct, reporting or dissemination of this study.

## RESULTS
COVID-19 presented staff in critical care with a rapidly changing situation and guidance; staff found themselves working in a state of constant flux. Participants' accounts reflected changes every shift, with one person saying every day felt like the first day in a new job. Below, we describe the features of working in this extreme pandemic context in relation to the seven features of a community of fate, highlighting not just changes in working practice but the social corollaries of these changes. Illustrations from the data are given in figures 2–5.

### Danger recognition: fear and dread of COVID-19
Many staff members commented on the anxiety they felt in anticipation of working in critical care during COVID-19. During the early stage of the pandemic, these anxieties were heightened by media reports of overwhelmed hospitals in Italy and exhausted healthcare workers in China. Staff were also acutely aware of their personal risk of catching COVID-19 and taking this home to their families. Particularly affected were those from ethnic minorities and those with at-risk and shielding family members. One black African nurse living in a multigenerational household described the impact the death of a fellow nurse from COVID-19 had had on her, and how her sister, also a nurse, had also fallen ill with COVID-19. Such experiences heightened the anxiety of working in critical care; they also entailed considerable emotional labour—something described by several respondents when talking about how they had sought to reassure their partners and children that they were safe at work. A number of participants had experienced the death of colleagues, which was deeply affecting.

### Moral density: purpose and duty
In spite of—indeed because of—this existential threat, many staff members spoke of a strong sense of duty in relation to working during COVID-19. Recognition of the danger the pandemic posed to the population as a whole was a powerful motivator, prompting several redeployed staff members in the sample to proactively volunteer. Shared professional commitment was also a powerful factor, with some expressing that this was simply their job, and others that it was what they had been trained to do. This collective vocation to provide care and be present created a common sense of purpose, which cut across professions and hierarchies.

**Danger recognition**

"I was terrified... I can't really remember my journeys going home, I think I cried, pretty much all of the time." (Critical Care Nurse)

"There probably was an existential angst in the background because you always thought, 'oh god, are they going to run out [of PPE]? Are we going to have to make one set last all day?'...I had a couple of sleepless nights thinking, 'oh, what's it going to be like tomorrow?'" (Redeployed Nurse)

**Moral Density**

"I just felt I had to do it...a lot of the people who were there who were going back in, lots of the staff were, I think, there was lots of talk of them all having written their wills ... you know, you're aware of the health workers in critical care who had died and had become ill and died in China and in Spain and also in London." (Redeployed Nurse)

"I think I'd have struggled...to know that I could have been there to help and didn't" (Redeployed Nurse)

**Closure**

"I felt more isolated from the rest of the hospital because it felt like people were afraid to come to ITU, so the staff were afraid to come to us. So, because of only doing night shifts, I didn't really see anybody on a higher level than me, I didn't see infection control come to see us, I didn't see head of nursing come to see us, from other units or from other areas. So it felt like we were very on our own, as a unit." (Critical Care Nurse)

"There were times that you came out at night after your shift, there wasn't a car on the road, there wasn't a shop open, it was just like you just thought, my goodness, there's nothing else out here but COVID." (Critical Care Nurse)

**Figure 2** Data extracts illustrating critical care as a community of fate during COVID-19. ITU, intensive therapy unit; PPE, personal protective equipment.

## Trial: ordeals in critical care

Working in COVID-19 critical care was extremely challenging. The main ordeals that staff described were as follows (see also figures 3 and 4):

### Dislocation

Setting up COVID-19 critical care facilities often involved converting clinical areas, including wards and theatre recovery areas, to new purposes and assimilating redeployed staff into newly assembled critical care teams. Adapting to these new circumstances proved challenging, with staff reporting difficulties locating equipment and supplies, or identifying who was in charge. Lack of familiarity with other team members was exacerbated by PPE, which rendered identification and recognition difficult. Redeployed staff without previous critical care experience faced particularly acute challenges adapting to unfamiliar language and processes, with some saying they felt ill equipped to deal with even basic tasks such as how to record observations or wash patients. This was particularly so for the operating department practitioners (ODPs) in our sample.

### Responsibility

Staff described a rapid acceleration in levels of responsibility. This included managing the dual tasks of caring for critically ill patients while also training non-critical care staff, which created additional cognitive and emotional demands: some senior nursing staff reported not taking a break for 6 or 7 hours due to anxiety over leaving inexperienced staff. These demands were magnified for staff from smaller units with a smaller pool of experienced staff to draw on for redeployment into critical care. Extreme stress was reported by senior nursing staff trying to maintain adequate staffing levels in these contexts. The sheer number of patients also exacerbated the burdens of responsibility. The decision to abandon existing guidance on minimum staff:patient ratios[25] was perceived to be unsafe by some nursing staff, and led to a loss of confidence, even for some experienced nurses. Some reported suffering from extreme anxiety and a sense of loss of control when attempting to look after patients safely and with dignity, while nurses in several units expressed sadness at their inability to provide as much care as usual in terms of washing, turning and personal care.

### Caring for patients

Staff commented repeatedly on the physical and emotional intensity of caring for critically ill patients with COVID-19, and their acute awareness of how frightening it was for patients. While the majority of staff felt safe and protected by their PPE, it nonetheless made caring for patients difficult due to loss of manual dexterity, numbing of the senses, loss of visual and audio cues, heat, weight, dehydration, facial pain and the fact that everyone looked the same. This also disrupted professional interaction: staff were sometimes unable to recognise who had the expertise to respond to a given request for help, and it hindered tasks that depended on coordinated teamwork such as proning or turning patients.

> **Trial**
>
> *Dislocation*
> "The theatre nurses were really very anxious and very upset, and there was a lot of time spent on their emotional wellbeing. So, sometimes they would leave the ward in tears, because they felt they couldn't cope, you know? It was very traumatic for everybody." (Critical Care Nurse)
>
> "There was no understanding or appreciation of how long I'd been out of Intensive Care, and also what my current arrangements were at this point in my life, and that doing a 12.5 hour shift was actually something that was quite difficult to do." (Redeployed Nurse)
>
> "I could have done with more training – you know, how the ventilator works, what settings it could be on, their charts, you know, their obs charts – they were hard to take on board in the first few days because there's so much to write on it, and understanding how important that chart is for seeing how the patient's getting on – that wasn't really addressed until we were actually in the unit and they realised that, oh, my goodness, these people don't know how to fill out these charts, we need to teach them." (Redeployed ODP)
>
> *Responsibility*
> "It wasn't safe to go to the toilet sometimes because the junior staff that were on, you couldn't leave. Hence why sometimes I didn't get lunch or breakfast for six, seven hours." (Critical Care Nurse)
>
> "So it was quite a difficult thing to be thrown into a team of people who didn't know each other, didn't know what skills everybody had in a very intense situation where people were extremely sick and at every point you had somebody to train, because we were constantly in a state of trying to train up people in case we had an increase in workload. So most of my shifts you were training somebody as well as taking care of a very sick person, as well as working with colleagues you didn't know, and you didn't know their skill sets, so it was quite a demanding situation." (Redeployed Nurse)
>
> "It was not care that I would ever want to associate myself with. I felt patients' care was just compromised because of…lack of resources and staff and everything. And it was just so demoralising to know that you weren't providing the care that you are more than capable of providing on a normal basis." (Critical Care Nurse)
>
> *Caring for Patients*
> "I pretty much fell apart on, on a very regular basis and I remember even having a panic attack in the middle of the unit, round there, because I had 4 haemofiltrations running and I was the only person in the entire place and we had 10 level 3's, three of which were proned, four were haemofiltered and I was the only person filter trained." (Critical Care Nurse)
>
> "There was one shift where I thought I was going to have a panic attack in the PPE… it's that sense of, I can't breathe in this, and then you have to try and rationalise it and say, no, you can, it's just very hot, it's very uncomfortable. But particularly at the beginning, full PPE was just…it was just awful, the weight of the visors that we had initially gave you a headache, a sore neck, and the breathing that I've mentioned." (Redeployed Nurse)
>
> "Your senses were quite numbed by the PPE. I remember there was one patient, they weren't intubated and their oxygen saturations were okay, so all the obs on their screen were okay, but actually it was only after I came into the bedspace that I actually realised the patient was in quite a lot of distress and I hadn't really noticed." (Doctor)
>
> "It was like 15 years of sick patients coming in, in a fortnight…And there was one of us between maybe four or five beds. And that stress was incredible. Because all you heard was, 'I need an ITU nurse', 'I need an ITU nurse', 'I need an ITU nurse'." (Critical Care Nurse)

**Figure 3** Data extracts illustrating critical care as a community of fate during COVID-19. ITU, intensive therapy unit; ODP, operating department practitioner; PPE, personal protective equipment.

The lack of an evidence base for treating COVID-19 also led to fundamental uncertainty about what care to provide. Medical and nursing staff alike commented on the simultaneous information vacuum and information deluge, compounded by the lack of a central, controlled source for information about clinical practice, with much communicated by word of mouth. AHPs described new challenges associated with 'caring at a distance', when direct access to patients was either not possible or limited. For example, one dietitian spoke of how relying on verbal reports from nursing staff via telephone rather than seeing the patient and their charts themselves made it difficult to assess nutritional status and ensure appropriate supplies.

### End of life
While many participants were used to caring for dying patients, COVID-19 brought new difficulties. With families unable to visit, staff's emotional relationship with patients was intensified, with many staff members saying

**Trial (Cont.)**

*End of Life*

"I don't think I've felt as prolonged a period of intermittent sadness and grief for my patients as I had during the pandemic. …it was because there were so many of them in such a short period in time. And, particularly extra bits of empathy that you have for their families, which you take on some of that emotional burden because their families aren't there with them. And that genuinely was the most distressing bit; and it did, and you felt physically sad and exhausted during those periods and times that you were comforting the relatives over the phone, the staff that were around them, yourself, your junior staff." (Doctor)

"I've dealt with death a lot over the years, but there was one day that I just…I thought, I can't…if I have to do that again, look after somebody who was dying, I don't know if I could do it, and I've never had that experience before, with 20 odd years of ITU… I've never thought, 'I just don't know if I can do that'. And that was a COVID specific thing, just because it had been so… challenging to be able to do it all remotely… to not be able to support the relatives…and just not have them there, it just didn't seem right at all." (Redeployed Nurse)

"Patients who did die, the mortuary porters weren't allowed in, so then we were dealing with loading bodies onto mortuary trolleys, which is not something I've ever had to do in my 20 years before… that was one of the most difficult things." (Redeployed Nurse)

"I got upset, like I remember we'd got this pile of notes, and we were going through to see where the patients had been moved to, and unfortunately, I think we had about a pile of ten, and six of them had died. And I can remember getting upset, thinking, oh my God, I can't believe that." (Redeployed Ward Clerk)

*Interaction with Families*

"The interaction with relatives was shocking. It was really distressing that the patient's relatives couldn't be with them… they're normally a very big part of our work in the ICU." (Doctor)

"I found my strategy of how to speak to families didn't work at all well during COVID. I had probably the two most dysfunctional, disastrous conversations I've ever had I think as a consultant, trying to speak to the partners of patients who were very sick in our intensive care" (Doctor)

"During COVID, unless they were dying, as in about to die, the relatives never came in. And that had far more of a toll than I think any of us ever expected. I consider myself quite a tough person. You know, I will cry occasionally if a patient dies, but this was…this wasn't like anything I've experienced before. This was…to have a family, there is so much that you read and do that is non-verbal in terms of communication that you clearly can't do when they are at the end of the phone. You can't read their emotions, you can't read their reactions. You can't comfort them, you can't touch them. All of that stuff, which makes us feel like we've done a good job, even though the patient has died, a lot of that stuff was again just completely foreign now. Because you are having to do it over the phone. This isn't something we have lots of experience at. This is something that we are feeling our way through and that was really distressing for lots and lots of us." (Doctor)

**Figure 4** Data extracts illustrating critical care as a community of fate during COVID-19. ICU, intensive care unit; ITU, intensive therapy unit.

they felt they had to take on the family's role of 'being there' for the patient. All the staff members we interviewed expressed deep sadness at witnessing the deaths of patients with COVID-19 who were not allowed to have a family member present in person. Staff who sat with patients at the end of life often found it heartbreaking to be party to this very intimate moment between a patient and their family, for example, while holding a telephone to the patient's ear. Staff reported that the decision to allow family members in again towards the end of life had made a huge difference and 'humanised the process again'.

Staff also encountered new challenges after a patient had died. Some nurses reported that protocols regarding what to do after a patient had died from COVID-19 were not clear at the start, for example, around last offices, infection control and what to do with patient belongings. One experienced nurse described how upsetting it was to be tasked with moving people who had died into body bags and onto trolleys for the morgue.

The severity of illness and high death rate in COVID-19 critical care, while difficult for all staff, was particularly hard for redeployed theatre and recovery staff, whose work usually involves patients who improve. The ODPs we spoke to reported having no training or experience in communicating with families at end of life. Ward clerks were also affected by the sheer numbers of deaths and caring for these patients' families. As the first port of call

**Material and organisational resources**

"So they all pitched in. We had surgeons coming to help us. We had orthopaedic surgeons coming to the teams to help us move ICU patients… We had anaesthetists coming and re-learning skills. We had a million and one staff who had previously worked in ICU coming back and saying, 'how can I help?' We were, genuinely at some points, inundated with offers of help." (Doctor)

"We set up an email for all the families to be able to …email in photographs or pictures for their family, and we could print them off, laminate them and hang them up by the patient. And that started to individualise the patients and bring their families together with them even though they couldn't be there in the flesh." (Redeployed Nurse)

"I think that being able to set up the video links really helped…being able to kind of show families their loved ones rather than just in words but in a visual representation made such a big difference." (Critical Care Nurse)

"Things that take literally years to change usually were getting changed within days, if not weeks, and that in itself for the NHS is a miracle, so that is definitely a positive." (Redeployed Nurse)

**Axis of convergence**

"Teamwork was one of the things where, actually, humanity, certainly in the hospital, really pulled together." (Doctor)

"So the physiotherapy team, my direct team that works within critical care, we were very proactive, I think we supported each other really nicely, you know, our manager was wonderful, she gave us whatever support we needed…you know, making sure that you were able to take annual leave if you needed it.  Just sitting and talking more, I feel like we did so much talking throughout the whole thing, I think that level of support and teamwork was wonderful." (Physiotherapist)

"I had a great sense of, you know, common purpose, which was, I'm sure, very good for my mental health…I think there were many positives to being at work in an intense environment with a common purpose" (Redeployed Nurse)

"The workplace, did for that brief time, transform into that kind of proactively caring organisation we would hope that it would be all the time." (Doctor)

"It was a really positive experience. I worked with some amazing staff who were incredible… I worked with people who have made me laugh, who gave such amazing care, who were so knowledgeable about their skill, their craft. Unbelievable. I was in such awe of these staff… and I can truthfully say I know I did the best I can. So, I'm very proud of myself." (Redeployed ODP)

"It was nice to see how the team came together and worked together and, you know, every member of the team, from domestics, to physios, to nurses, to ACCPs, to the junior doctors, to the consultants – I think everybody was there, and you felt that everybody was there, which was nice." (Critical Care Nurse)

**Social ritual**

"We went in pairs, we made sure that we always went with somebody else, and I think that was really good just for moral support. And we made sure that we left together so that we had somebody else to help make sure we were doing our PPE correctly" (Dietician)

"We would do our best to keep spirits up… I had my hair dyed kind of blue … NHS blue as it were. So, people could identify me and they knew it was me, but on the front of my top, rather than writing [my name] … I would put things like 'I am Pac Man' or, 'La-La from the Teletubbies', or something, just to kind of keep it…a slight element of levity." (Doctor)

"I used to draw flowers on my gown and just little things … or draw silly faces on your visor." (Redeployed nurse)

"If you come home and tell family that you had to wear PPE, you know, until they've actually donned the stuff, I think, and doffed it, for that matter, they don't really get it" (Redeployed nurse)

**Figure 5**  Data extracts illustrating critical care as a community of fate during COVID-19. ACCPs, advanced critical care practitioners; ICU, intensive care unit; NHS, National Health Service; ODP, operating department practitioner; PPE, personal protective equipment.

for families phoning up, this work could be very emotionally intense; as one ward clerk observed, 'sometimes you could just listen to them, even though you couldn't help them'.

### Interaction with families

Caring for families is a large part of critical care work,[26 27] and particularly important at the end of a patient's life.[28 29] With visitors excluded from patients' bedsides due to COVID-19, staff experienced additional demands to keep families informed while navigating the constraints of communicating 'virtually' using digital technologies/telephone. Staff spoke of the peculiar difficulties of avoiding unwarranted optimism or pessimism when families were unable to witness for themselves how a patient might be progressing or deteriorating. The need to communicate at a distance also made it particularly challenging for staff to break bad news. Most staff felt unprepared to have these conversations, and consultants,

in particular, reported having some of the most upsetting conversations of their careers.

The emotional strain of facing these trials, combined with the sense of isolation, was often severe. Some nursing staff reported experiencing acute fear, stress, anxiety, exhaustion and burnout, particularly in smaller units. Staff spoke of crying on the way to work, breaking down in tears on shift and crying after leaving work. Fear for the consequences of what they perceived to be inadequate staffing levels, inexperienced staff and a high volume of critically ill patients left some nursing staff feeling 'broken'. Across the sample, staff reported a range of negative impacts, such as sleep disturbance, panic attacks, weight loss/gain, and feelings of guilt, grief, anger, sadness and dread. Some accessed professional mental health services, either within the Trust, through their general practitioner or privately.

### Closure: isolation and the 'COVID-19 bubble'
With COVID-19 critical care facilities physically and socially isolated from other parts of the hospital for infection control purposes, and many staff members removed or redeployed from their usual workplaces and colleagues, work in what some staff referred to as the 'COVID-19 bubble' could feel like a collective exile. Several participants voiced disappointment that COVID-19 seemed to be treated as 'a problem for critical care' rather than the hospital at large, such was their sense of isolation. This was magnified for staff working night shifts, who commented on feeling forgotten, having less food and drink available, fewer redeployed staff to support them and less visibility of senior managers.

### Material and organisational resources: learning and creativity
By contrast, staff also took pride in the many ways they had managed to adapt to the challenges posed by new working arrangements. Staff experienced a steep learning curve: consultants described the challenges of treating patients in the absence of an evidence base, while nurses spoke of learning to manage a large number of patients who were considerably sicker than usual and whose condition could deteriorate quickly. Both groups spoke of the quick, self-directed learning needed to stay abreast of rapidly changing treatment protocols. While some felt that 'nothing prepares you to work in a pandemic', those with experience of working in previous infectious disease outbreaks such as SARS or H1N1, as well as those involved in protocol development and training for emergency situations, felt better prepared.

Rapid learning across units was helped by the relative homogeneity of the cohort of patients with COVID-19. The following training was specifically mentioned as helpful: locally run, structured competency and skills training; the FutureLearn COVID-19 Critical Care course; and the frequent Intensive Care Society webinars. For nursing staff, much of the training occurred on the job; those who had been redeployed particularly valued the opportunity to shadow staff and/or to have a more experienced buddy. Staff across nursing and medical teams identified platforms like WhatsApp as a crucial means of sharing information and updates, moving away from traditional modes of communication.

There was widespread praise for the speed with which new systems had been put in place and change effected in the NHS. A particular success was the use of tablets and mobile phones to connect families and patients via synchronous/asynchronous video-conferencing. Staff observed that the more frequently they were able to connect with families, the easier this virtual relationship became, the more patients were individuated, and the more satisfactory the caring relationship.

### Axis of convergence: teamwork
Almost everyone we interviewed articulated positive aspects of their experience during the pandemic. Foremost among these was the sense of teamwork and camaraderie that had developed as staff pulled together. Many felt proud to have been part of the pandemic response, and spoke of the satisfaction of being part of something and working for a common purpose, while the influx of redeployed staff was often felt to be a source of both moral and practical support. Newly qualified and experienced nurses alike said that working during COVID-19 reaffirmed the values that had taken them into the profession in the first place.

Teamwork, and the mutual support it provided, also characterised some of the measures that proved most effective in meeting the challenges of COVID-19. Daily team huddles were observed to be useful for identifying and trouble-shooting local issues, while shared conversations within the team about difficult shifts and patients who had died was helpful in coping with emotionally difficult experiences. As a result, various staff said they had gained improved clinical, operational and management skills, increased resilience, confidence and self-esteem.

### Social rituals: donning and doffing
In a setting that depends so heavily on teamwork, it would be futile to try and draw any hard-and-fast distinction between routine and ritual: both serve at once to provide reassurance and to affirm a common identity in the face of disorder and danger. In the context of COVID-19, however, some routines acquired special significance. The donning and doffing of PPE was one such. Staff spoke of the benefits of going into critical care in pairs to check both on PPE fit and each other's well-being. They also found creative ways to decorate their PPE, turning it from a faceless signifier of risk into an expression of individuality. In so doing, they transfigured the fear of entering critical care into a moment of human solidarity and interconnectedness.

### DISCUSSION
Using the sociological concept of 'community of fate', our analysis shows how working practice in critical care

changed during the first wave of the COVID-19 pandemic in the UK, and how staff mobilised their collective resources to provide care to patients. In Baehr's Weberian use of the term, 'community of fate' does not imply fatalism. On the contrary, it denotes the condition of purposeful collective action that may be attained by a group of people facing a common crisis. Employing the concept to analyse our interviewees' testimony not only helps explain staff experience in the face of extremity, but highlights the crucial role of solidarity and teamwork in achieving a functioning COVID-19 critical care system.

That collective achievement should not blind us to the anguish that many participants endured and its lasting damage: several staff expressed a deep reluctance to return to critical care to tackle a second wave, something highlighted in other qualitative studies of frontline staff experience.[30] Nor should we assume that the communities of fate that coalesced in response to the first wave will survive as the pandemic becomes increasingly protracted and challenges recur. As Baehr notes "Where all hope is gone, resources spent, and action deemed hopeless, communities of fate are impossible."[18] It therefore behoves us to ensure that resources *are* available and the conditions for good care and staff well-being are optimised. While some have suggested this should focus on *individual* well-being initiatives, including mindfulness and other coping strategies,[31] our analysis underscores the importance of *structural* resilience in critical care and attending to the conditions under which teams can prosper.

Our study is not alone in emphasising the importance of taking healthcare workers' experiences into account during the COVID-19 pandemic. Indeed, our findings echo and amplify many of those from the Rapid Research, Evaluation and Appraisal Lab Study of the perceptions and experiences of healthcare workers during COVID-19.[6 16 32] While the latter drew on a sample consisting primarily of doctors, our study adds the voices of nurses and other professionals to the evidence base, as well as extending its geographical reach beyond London to other parts of the UK. By drawing on sociological theory to interpret our data, we provide an alternative lens through which to understand social cohesion and transformation in critical care during pandemic times.

Some limitations to our study apply, namely the self-selecting sample, the small number of sites from which we recruited and the small number of AHPs. As such, our findings may over-represent the experiences of nursing and medical staff, and further research should consider a broader range of experiences from across the professions. While we recruited from a range of critical care settings across the UK, these were all located in urban areas and staff working in rural areas may have different experiences. Nonetheless, we believe the concept of 'communities of fate' is likely to have broad theoretical generalisability. Finally, the rapid analytical methods we used were designed to structure the analysis thematically, and were not explicitly oriented to exploring differences by demographic variables, such as gender and ethnicity.

Studies of staff experience during COVID-19 have shown how these variables shape experiences;[32 33] we acknowledge the importance of recognising the 'stratified forms of risks and vulnerabilities facing diverse groups of healthcare workers both within and across health systems'.[34]

A key strength of this study is its in-depth focus on critical care during the first wave of the COVID-19 pandemic in the UK. Our data have alerted us to a range of specific measures that might be implemented at the local level to help critical care staff contend with the challenges posed by COVID-19, and we have made our recommendations available and accessible.[35] They join a growing body of guidance and resources aimed at helping staff maintain mental health, well-being and resilience through the pandemic.[36 37] As has been noted elsewhere,[16] with some notable exceptions,[38] these resources are overwhelmingly oriented towards supporting individuals. Yet as Rose *et al*[15] argue, individualised psychology-based interventions will be ineffective unless the social preconditions for well-being are in place. Our own findings strongly support that view. While personalised support is to be welcomed, attention also needs to be paid to the social and organisational conditions in which individuals work.

Our research shows the importance, on the one hand, of building and facilitating teamwork within and across critical care; and on the other hand, of addressing the sense of isolation that critical care staff felt from other parts of the healthcare system. Resourcing is one aspect of this: our data attest to the need to address anxieties around practical issues such as staffing levels and PPE availability. But equally important is how institutions and national bodies develop transparent plans to deal with COVID-19 and meaningfully engage with frontline staff. Responsibility is key, and time, energy and resource must underpin the professional duty of care to healthcare colleagues in order to comprehensively manage surge situations like the COVID-19 pandemic.

## CONCLUSION

To the best of our knowledge, this is the first sociological analysis of healthcare staff's experiences of working in critical care during the first wave of the COVID-19 pandemic in the UK. Our findings provide timely insight into the challenges of critical care work during the first wave of COVID-19 and suggest the importance of moving beyond an individualised understanding of staff well-being to consider the social and organisational factors at stake.

**Acknowledgements** We would like to thank all the NHS staff who took part in this study for giving up their time and sharing their experiences with us for this research.

**Contributors** CMM and ABD conceived the study, and all authors contributed to the study design. CMM, SH and NP collected the data. CMM, SH, CMc, SS and NP analysed the data and all authors contributed to data interpretation. CMM, SS and

NP wrote the first draft of the article and all authors contributed to subsequent revisions. All authors had full access to the data.

**Funding** This research was funded by Medical Research Scotland through a COVID-19 Research Grant (CVG-1739-2020) and supported in part by the Wellcome Trust (209519/Z/17/Z).

**Disclaimer** The funders of the study had no role in study design, data collection, data analysis, data interpretation or writing of the manuscript.

**Competing interests** None declared.

**Patient consent for publication** Not required.

**Ethics approval** Ethical approval was granted by the University of Edinburgh School of Social and Political Science Research Ethics Committee; HRA approval (20/HRA/3270) was also obtained.

**Provenance and peer review** Not commissioned; externally peer reviewed.

**Data availability statement** The datasets used and analysed during the current study are available from the corresponding author on reasonable request.

**ORCID iDs**
Catherine M Montgomery http://orcid.org/0000-0002-5829-6137
Sally Humphreys http://orcid.org/0000-0002-4397-6404
Steve Sturdy http://orcid.org/0000-0002-3273-1727
Natalie Pattison http://orcid.org/0000-0002-6771-8733

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
