## [Reviewer comments · BMJ Open]

ARTICLE DETAILS

TITLE (PROVISIONAL)	Critical care work during COVID-19: A qualitative study of staff experiences in the UK
AUTHORS	Montgomery, Catherine; Humphreys, Sally; McCulloch, Corrienne; Docherty, Annemarie; Sturdy, Steve; Pattison, Natalie

VERSION 1 – REVIEW

REVIEWER	Braquehais, María Dolores Galatea Clinic. Galatea Foundation, Inpatient Unit. Integral Care Program for Sick Health Professionals
REVIEW RETURNED	28-Jan-2021

GENERAL COMMENTS	I want to congratulate the authors for their novel contribution to the discussion of the impact of the pandemics on healthcare workers from a new sociological perspective that emphasizes the importance of "community of fate" group response. I only wanted to make some minor comments: 1) I suggest that the 2 last key messages should be identified as a weakness of the study.2) With regards to the methodology section,2.1. There should be a clearer description of the sample recruitment method (potential respondents, respondents per hospital, etc). Maybe it could be summarized in a figure.2.2. It is also important to describe each hospital characteristics and more information on the COVID epidemiological and clinical status of each hospital during the studied period (i.e: responses may vary depending on several hospital variables; urban/ rural; personal equipment availability; ICU overload; numbers of COVID patients/ total beds of the hospital, etc).2.3. It is also important to describe when the study was conducted as the responses to the COVID pandemics have varied. There may be significant differences between the first and following waves' responses.3) Some of these concerns (see point 2) may be reflected in the discussion section.
---

REVIEWER	Vindrola-Padros, Cecilia University College London, Department of Targeted Intervention
REVIEW RETURNED	09-Feb-2021

GENERAL COMMENTS	This is a very interesting and well-developed manuscript. I hope the authors find the following suggestions helpful when making changes. -the manuscript only requires minor changes before it is suitable for publication. The authors indicate that one of the reasons why
--

	the study was different to other qualitative studies exploring the experiences of healthcare workers delivering care during the pandemic in the UK was due to their use of a diverse sample of HCWs. I was therefore expecting a deeper reflection on how the experiences of HCWs in their sample varied by professional group or perhaps other variables identified by the authors. Some recent work has been published in the UK exploring how gender and ethnicity have shaped these experiences, so it would be good for the authors to reflect on these issues in more detail in the results and discussion. -related to this point, the authors indicate that they used a rapid analysis approach using structured templates to facilitate deductive/inductive analysis. Did any of these templates structure data analysis to facilitate the comparison of experiences between prof groups or to capture variation within the sample? -when reflecting on the limitations of the study, it would be good to know if the authors identified any limitations in relation to the rapid analysis approach, the use of structured templates and the use of a matrix to organise the data. -due to the rapidly evolving evidence base on COVID-19, I would advise the authors to run a new search of the literature before resubmitting the manuscript to make sure the references are up to date. A few papers on qualitative studies focusing on the experiences of HCWs in the UK have been published recently in this journal and others and these are not cited in the manuscript.
--	--

VERSION 1 – AUTHOR RESPONSE

Reviewer 1 comments	Response
I want to congratulate the authors for their novel contribution to the discussion of the impact of the pandemics on healthcare workers from a new sociological perspective that emphasizes the importance of "community of fate" group response.	Thank you
1. I suggest that the 2 last key messages should be identified as a weakness of the study.	We have rephrased these last two points to more clearly emphasise them as limitations of the study.
2. With regards to the methodology section, 2.1. There should be a clearer description of the sample recruitment method (potential respondents, respondents per hospital, etc). Maybe it could be summarized in a figure.	We have provided the number of respondents recruited from each hospital.
2.2. It is also important to describe each hospital characteristics and more information on the COVID epidemiological and clinical status of each hospital during the studied period (i.e: responses may vary depending on several hospital variables; urban/ rural; personal equipment availability; ICU overload; numbers of COVID patients/ total beds of the hospital, etc).	We have provided some further contextual information about the hospitals under 'Sampling and Recruitment', including the size of the critical care units from which we sampled. We did not have approval to collect other data on e.g. numbers of COVID patients or PPE availability and as this information is not in the public domain, we are unable to provide it here. Doing so would

	also risk making the sites identifiable, which would contravene the terms of our ethics approval.
2.3. It is also important to describe when the study was conducted as the responses to the COVID pandemics have varied. There may be significant differences between the first and following waves' responses.	We describe in the Methods that interviews were conducted between Aug and Oct 2020. We have now added that semi-structured interviews covered staff's experiences of working in critical care during the first wave of the pandemic (roughly between March and July 2020).
3) Some of these concerns (see point 2) may be reflected in the discussion section.	As we have been unable to go into detail about the epidemiological and clinical context of the hospitals from which we recruited, we have not been able to reflect on this in the discussion. However, we have noted in the limitations that given that we only recruited from urban areas, findings may differ in rural settings. We believe the concept of 'communities of fate' is likely to have broad theoretical generalisability.
Reviewer 2 Comments	Response
This is a very interesting and well-developed manuscript. I hope the authors find the following suggestions helpful when making changes. The manuscript only requires minor changes before it is suitable for publication.	Thank you for these constructive suggestions, which have helped us strengthen the paper.
The authors indicate that one of the reasons why the study was different to other qualitative studies exploring the experiences of healthcare workers delivering care during the pandemic in the UK was due to their use of a diverse sample of HCWs. I was therefore expecting a deeper reflection on how the experiences of HCWs in their sample varied by professional group or perhaps other variables identified by the authors. Some recent work has been published in the UK exploring how gender and ethnicity have shaped these experiences, so it would be good for the authors to reflect on these issues in more detail in the results and discussion	While reports of some other studies have focused predominantly on doctors, our sample comprised a much higher proportion of nursing staff, and this is where we see important added value in our work. Throughout the manuscript, we specify where findings relate to particular staff groups. To strengthen this contribution, we have added more examples from ODPs, ward clerks and allied health professionals. In relation to gender and ethnicity, neither our topic guide nor our analysis were specifically designed to interrogate these aspects of experience. We now explicitly reference this as a limitation of the study and cite those studies focusing their analysis on these aspects of experience.
Related to this point, the authors indicate that they used a rapid analysis approach using structured templates to facilitate deductive/inductive analysis. Did any of these templates structure data analysis to facilitate the comparison of experiences between prof	The matrices were designed to structure the analysis thematically, and were not specifically oriented to exploring differences by demographic variables, such as gender and ethnicity. Variation within the sample was explored through team discussion of how the

groups or to capture variation within the sample?	results varied by professional group – information which was captured on the initial summary templates.
When reflecting on the limitations of the study, it would be good to know if the authors identified any limitations in relation to the rapid analysis approach, the use of structured templates and the use of a matrix to organise the data.	In relation to above, we now reflect on this as a limitation of the study, as indeed it would have facilitated the analysis to be able to visualise the data across multiple dimensions.
Due to the rapidly evolving evidence base on COVID-19, I would advise the authors to run a new search of the literature before resubmitting the manuscript to make sure the references are up to date. A few papers on qualitative studies focusing on the experiences of HCWs in the UK have been published recently in this journal and others and these are not cited in the manuscript.	Thank you for pointing us in the direction of this further literature, which we now reflect on and reference in the Introduction and Discussion.